# REPEATED INTEGER LINEAR PROGRAMMING FOR BIT SELECTION IN NEURAL NETWORK QUANTIZATION

## ABSTRACT

Network quantization methods, which have been widely studied to reduce model size and computational cost, are now becoming well established as practical solutions. Mixed-precision quantization, which assigns optimal bit widths to layers, blocks, or other substructures, offers a promising approach to balance model performance and efficiency. However, determining the optimal bit configuration is a challenging combinatorial optimization problem, as it requires selecting discrete bit widths for multiple substructures across the network. In this paper, we propose an efficient algorithm that approximates the problem as an integer linear program and iteratively explores the bit-configuration space. Our method utilizes a small set of unlabeled samples with a low computational overhead, making it compatible with both widely adopted quantization methods: post-training quantization and quantization-aware training. We demonstrate the effectiveness of our approach in both settings, consistently achieving superior performance compared to single-precision baselines and existing bit-selection methods. The code will be released upon acceptance.

## 1 INTRODUCTION

In recent years, deep neural networks (DNNs) have achieved remarkable success across a wide range of tasks, including image recognition (Dosovitskiy et al., 2020; He et al., 2016a; Tan & Le, 2019), object detection (Redmon et al., 2016; Ren et al., 2015), speech processing (van den Oord et al., 2016), and natural language understanding (Vaswani et al., 2017). However, this success comes at a significant cost: state-of-the-art models often require substantial memory footprint and computational resources, making their deployment on low-end edge devices, such as mobile phones, embedded systems, and IoT devices, extremely challenging due to constraints on power, memory, and latency. To enable practical deployment in such environments, model compression has become indispensable.

Among various compression techniques, quantization—which replaces floating-point weights and activations with lower-precision integer representations—has emerged as one of the most hardware-friendly approaches. Numerous quantization methods have been proposed, including both uniform (Jacob et al., 2018; Wei et al., 2022) and non-uniform (Li et al., 2020; Gongyo et al., 2024) schemes, to reduce model size and computational overhead while preserving accuracy.

Quantization approaches can be broadly categorized into quantization-aware training (QAT) (Jacob et al., 2018; Esser et al., 2019; Bhalgat et al., 2020) and post-training quantization (PTQ) (Nagel et al., 2020; Li et al., 2021; Wei et al., 2022). QAT performs end-to-end retraining by using many labeled data. To circumvent the non-differentiability of quantizers, the retraining often relies on the Straight-Through Estimator (STE) (Bengio et al., 2013), which allows gradients to pass through quantization operations unchanged during backpropagation. Although it achieves outstanding accuracy even at very low bit-widths (e.g., 4-bit), it is computationally expensive and time-consuming. In contrast, PTQ uses only a small amount of unlabeled data to calibrate the quantization parameters such as step size and threshold, and thus PTQ is considered as a more practical solution in hardware deployment, but often sacrifices accuracy when applied at lower precisions.

Despite advances in both QAT and PTQ methods, most existing studies (Esser et al., 2019; Liu et al., 2023) assign a uniform bit-width to all layers, with the common exception that the first and last layers

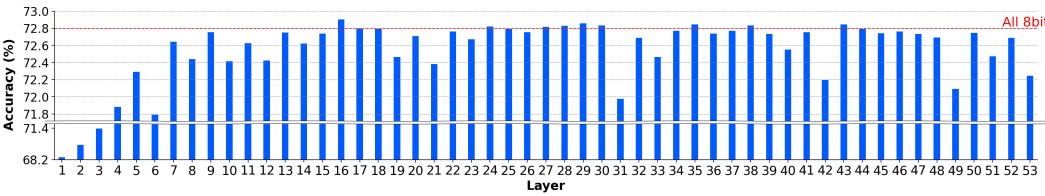

Figure 1: Sensitivity of quantization on each layer on MobileNetV2. Each layer is quantized to 3 bits, while others are set to 8 bits.

are heuristically given higher precision (e.g., 8-bit). However, the sensitivity to quantization varies significantly between layers. In Fig. 1, we visualize the sensitivity in the MobileNetV2 by assigning 8 bit in all layers and changing only a single layer to 3 bit. We can see that the resulting accuracy degradation differs significantly between layers. This observation suggests that selective bit width allocation per layer or substructure level, rather than uniform assignment, could lead to further improvements in the accuracy-efficiency trade-off. In fact, the widespread practice of assigning higher precision to the first and last layers implicitly acknowledges this non-uniform sensitivity and further motivates the development of bit-selection strategies.

Bit selection, however, is inherently a combinatorial optimization problem. For instance, assigning 4-bit or 8-bit precision to each layer in ResNet-18 (He et al., 2016b) results in $2^{18}$ possible configurations, each requiring a loss evaluation, which makes exhaustive search computationally infeasible. To address this challenge, we propose a novel method that formulates the bit selection problem as a sequence of integer linear programming (ILP) problems. By iteratively solving these relaxed optimization steps, our approach efficiently identifies high-performing bit allocations under quantization constraints.

Our main contributions are summarized as follows.

- Formulation of Repeated ILP for Bit Selection (RIBS): We formulate bit selection as an iterative ILP framework, making it computationally feasible.

- Theoretical connection between the original optimization problem and RIBS: We establish conditions under which the minimizers of the original optimization problem coincide with those of the ILP.

- Effectiveness of random block update and integration with reconstruction in RIBS: To mitigate the gap between the original optimization and its ILP approximation, we introduce random updates with a limited block size, and empirically identify optimal sizes for ResNet-18, MobileNetV2, and DeiT-T. We also propose a variant of RIBS integrated with PTQ methods that include reconstruction.

- Comprehensive evaluation in both PTQ and QAT: We validate RIBS on ResNet-18, MobileNetV2, and DeiT-T using the ImageNet dataset under diverse model size and BOP constraints. Across all settings, RIBS consistently achieves the highest accuracy among state-of-the-art (SOTA) single-precision and mixed-precision methods.

## 2 RELATED WORK

**Quantization-Aware Training (QAT).** Since the development of STE, numerous QAT techniques have been developed to enable end-to-end training of quantized neural networks. One particularly successful line of work involves jointly learning the quantization step size along with the network weights during training (Choi et al., 2018; Esser et al., 2019; Liu et al., 2022; Nagel et al., 2022; Gongyo et al., 2024). These methods allow the model to dynamically adapt its quantization scale to minimize task loss, significantly improving performance, especially under low-bit below 4-bit layer-wise settings.

**Post-Training Quantization (PTQ).** PTQ methods quantize pre-trained models using a small amount of unlabeled data while aiming to preserve generalization. There are two main approaches in PTQ. The first approach focuses only on determining step sizes (or threshold values). Initially based

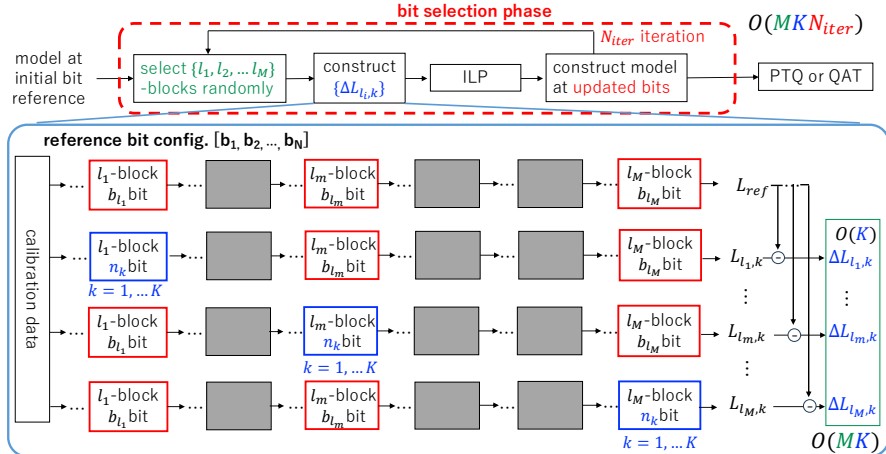

Figure 2: Overview of our method.

on min–max quantization (Krishnamoorthi, 2018), recent work has adopted MSE-based optimization for step sizes (Li et al., 2021).This approach has been further developed in ViT architectures(Li et al., 2023; Wu et al., 2024), combined with techniques such as reparameterizing channel-wise to layer-wise quantization after LayerNorm and applying $\log\sqrt{2}$ quantization to softmax activations.

The second approach is the reconstruction method that sequentially optimize the loss layer by layer or block by block after the determination of step sizes (Nagel et al. (2020), Wu et al. (2025)). Although more computationally demanding than step-size–only methods, reconstruction consistently achieves higher accuracy and is particularly well established in CNNs. Representative methods include Adaround (Nagel et al., 2020), which locally adjusts quantized weights by deciding whether to round up or down; BRECQ (Li et al., 2021), which shifts the optimization unit from individual layers to blocks; and QDrop (Wei et al., 2022), which introduces stochastic blending of floating-point activations during calibration to improve robustness. Collectively, these approaches have significantly advanced the state of PTQ in low-bit regimes.

**Mixed Precision.** Bit-width selection for quantized networks is inherently a combinatorial optimization problem, where direct search leads to an exponential explosion of configurations and is computationally infeasible. As a result, two main categories of alternative approaches have emerged: metric-based search and optimization-based methods.

Metric-based methods assign importance scores to layers using criteria such as Hessian spectra (Yao et al., 2021) or the degree of orthogonality in weights (Ma et al., 2023a). These scores are then used to rank layers and allocate precision accordingly. These are efficient but rely on proxies rather than true task loss, introducing approximation bias , even when adaptively selecting metrics (Dong et al., 2023).

Optimization-based methods reduce this bias by modeling quantization impact directly through reinforcement learning (Wu et al., 2018; Wang et al., 2019) and differentiable architecture parameters (Uhlich et al., 2019; Yang & Jin, 2021). However, they often require large datasets and heavy computation due to the exponential search space.

In contrast, we propose a novel optimization-based method that uses only a small amount of unlabeled calibration data, as in PTQ, and avoids computational bottlenecks by iteratively solving ILP problems. Using the distillation loss as the task loss, our method efficiently approximates the optimal bit configuration without relying on large-scale search or labeled data, bridging the gap between performance and efficiency.

## 3 METHOD

In this section, we give our formalism for computing the optimal bit configuration by iteratively solving the integer liner problem.

## 3.1 Preliminaries

Consider a neural network with $N$ substructures such as layers and blocks. To reduce resource consumption such as the latency and power usage while preserving high accuracy on edge devices, we quantize each substructure by assigning a bitwidth $b_i \in \mathcal{B}$ to each substructure $i \in \{1, \ldots, N\}$, where $\mathcal{B} = \{n^{(1)}, n^{(2)}, \ldots, n^{(K)}\}$ is a discrete set of $K$ allowable bitwidths (e.g., $\mathcal{B} = \{2, 3, \ldots, 8\}$). Let $\mathbf{b} = [b_1, \ldots, b_N]$ denote the vector of bit assignments across all substructures.

To measure efficiency of a mixed precision network, we use two constraints: (i) model size and (ii) bit operations (BOPs), defined as $C(\mathbf{b}) = \sum_{i=1}^{N} b_{w_i} b_{a_i} \mathrm{MAC}_i$, where $\mathrm{MAC}_i$ denotes the number of multiply-accumulate (MAC) operations in the $i$-th substructure, and $b_{w_i}$ and $b_{a_i}$ denote the bit-widths of the weights and activations, respectively.

The optimal bit configuration is obtained by minimizing the task loss $\mathcal{L}(\mathbf{b})$ under a resource constraint $C \leq C_{\max}$:

$$\min_{\mathbf{b} \in \mathcal{B}^N} \quad \mathcal{L}(\mathbf{b}) \text{ subject to } \quad C(\mathbf{b}) \leq C_{\max}. \tag{1}$$

However, this optimization yields a combinatorial optimization problem, where each candidate configuration requires evaluating the loss function. This makes the optimization process computationally expensive and challenging to solve directly.

## 3.2 Integer Linear Programming (ILP) problem

To avoid exhaustive search in equation 1, we first approximate the bitwidth-selection problem as a minimization of the sum of the task loss difference relative to a reference bit configuration $\mathbf{b}^0 = [b_1^0, \ldots, b_N^0]$.

$$\min_{\mathbf{b} \in \mathcal{B}^N} \quad \sum_{i=1}^{N} \Delta\mathcal{L}(b_i) \quad \text{subject to} \quad C(\mathbf{b}) \leq C_{\max}, \tag{2}$$

where $\Delta\mathcal{L}(b_i)$ denotes the loss increase when changing only the $i$-th substructure's bit-width from $b_i^0$ to $b_i$: $\Delta\mathcal{L}(b_i) \equiv \mathcal{L}(\mathbf{b}_{\hat{i}}^0) - \mathcal{L}(\mathbf{b}^0)$ with $\mathbf{b}_{\hat{i}}^0 \equiv [b_1^0, \ldots, b_{i-1}^0, b_i, b_{i+1}^0, \ldots, b_N^0]$.

This can be reformulated as a binary ILP by introducing binary decision variables $x_{i,k} \in 0, 1$, indicating whether bit-width $n^{(k)}$ is assigned to $i$-th substructure:

$$\min_{\{x_{i,k}\}} \quad \sum_{i=1}^{N} \sum_{k=1}^{K} \Delta\mathcal{L}_{i,k} \cdot x_{i,k} \tag{3}$$

subject to

$$\sum_{k=1}^{K} x_{i,k} = 1, \quad \forall i \in 1, \ldots, N, \qquad \sum_{i=1}^{N} \sum_{k=1}^{K} c_{i,k} \cdot x_{i,k} \leq C_{\max}, \quad x_{i,k} \in 0, 1 \quad \forall i, k, \tag{4}$$

where $\Delta\mathcal{L}_{i,k} \equiv \Delta\mathcal{L}(b_i = n^{(k)})$, and $c_{i,k}$ denotes the resource cost (i.e., BOPs or memory) of assigning $n^{(k)}$ to layer $i$. The first constraint ensures that each substructure is assigned exactly one bit-width. The second constraint enforces the resource budget. Since both the objective and the constraints are linear in the decision variables $\{x_{i,k}\}$, this problem can be efficiently solved using standard ILP solvers.

The original combinatorial problem in equation 1 and its ILP relaxation in equation 2 are related as follows.

**Theorem 1** *Let $\delta > 0$ be a fixed constant and let $\mathbf{b}^*$ be the optimal solution for the ILP in equation 3. Suppose the following two conditions hold for all $\mathbf{b} \in \mathcal{B}$:*

$$\left| \mathcal{L}(\mathbf{b}) - \sum_{i=1}^{N} \mathcal{L}(\mathbf{b}_{\hat{i}}^0) - \left( \mathcal{L}(\mathbf{b}^*) - \sum_{i=1}^{N} \mathcal{L}(\mathbf{b}_{\hat{i}}^{*0}) \right) \right| \leq \delta,$$

$$\sum_{i=1}^{N} \Delta\mathcal{L}(b_i) \geq \min_{\mathbf{b} \in \mathcal{B}^N} \sum_{i=1}^{N} \Delta\mathcal{L}(b_i) + 2\delta \quad \text{for} \quad \mathbf{b} \neq \mathbf{b}^*.$$

*Then the minimizers of the original and ILP losses coincide:*

$$b^* = \arg\min_{\mathbf{b}\in\mathcal{B}^N} \mathcal{L}(\mathbf{b}). \tag{5}$$

From this theorem the following two corollaries can be derived, which are more intuitive.

**Corollary 2 (Independent losses)** *Under the assumption that the bit-width of each substructure affects only its own loss contribution, i.e., the loss decomposes as $\mathcal{L}(\mathbf{b}) = \sum_i L_i(b_i)$ with the loss functions per substructure $L_i(b_i)$ then $\min_{\mathbf{b}\in\mathcal{B}^N}\mathcal{L}(\mathbf{b}) = \min_{\mathbf{b}\in\mathcal{B}^N}\left[\sum_{i=1}^N \Delta\mathcal{L}(b_i) + \mathcal{L}(\mathbf{b^0})\right]$, and equation 5 holds.*

This implies that when inter-substructure correlations are weak, the mismatch between the ILP solution and the true minimizer is small. In contrast, strong correlations often lead to substantial mismatches. As shown in our ablation studies, this effect is particularly pronounced in PTQ with reconstruction. The next corollary suggests another strategy to reduce such mismatches.

**Corollary 3 (Local neighborhood)** *Under the assumption that the search space $\mathcal{B}^N$ is restricted to configurations satisfying $d_H(\mathbf{b}, \mathbf{b}^0) = 1$, where $d_H$ denotes the Hamming distance, equation 5 holds.*

Corollary 3 shows that equation 5 is guaranteed when the bit-width configuration is updated by only a single substructure (i.e., within a Hamming distance of 1). However, such strictly local updates are inefficient: they require many iterations to make progress and are prone to being trapped in poor local minima. This situation is analogous to gradient descent with a very small learning rate—loss reduction is guaranteed, but convergence is slow and the risk of stagnation in local minima is high. By analogy, just as gradient descent benefits from a moderately large learning rate, our framework favors using a larger update size $M$ to accelerate progress and escape poor local solutions.

### 3.3 REPEATED BLOCK-WISE ILP

To address these issues, we propose an iterative ILP-based optimization, as outlined in Alg.1. At each step, the ILP is solved within a Hamming distance constraint of $M(> 1)$ from an updated reference configuration, allowing broader, yet controlled exploration of the solution space and reducing the risk of mismatching the optimal configuration. The ILP can be solved with equation 3 with the following constraint:

$$\sum_{k=1}^{K} x_{i,k} \le 1, \quad \forall i \in \{1,\dots,N\}, \tag{6}$$

$$\sum_{i=1}^{N}\sum_{i=1}^{K} x_{i,k} \le M, \ \sum_{i=1}^{N}\sum_{k=1}^{K} c_{i,k}\cdot x_{i,k} \le C_{\max}, \quad x_{i,k}\in\{0,1\} \quad \forall i,k, \tag{7}$$

Instead of directly solving this problem, we randomly select $M$ substructures from the total $N$ substructures at each step, and solve equation 3 subject to the constraints equation 4, with $N$ replaced by $M$. After $N_{iter}$ step, we adopt the bit configuration with the best task loss.

Moreover, to mitigate the effects of strong inter-layer correlations, we adopt a block-wise update strategy, where each block corresponds to a standard substructure in the network—such as a Basic Block in ResNet, an Inverted Residual Block in MobileNet, or an Attention Block in ViT. In this formulation, the block serves as the unit of bit selection, and all layers within a block share the same bit-width.

**Update of Step size** Quantization at $b$ bit inherently involves the determination of the step sizes or threshold values, which directly affect the performance of the model. Therefore, when evaluating the impact of changing the bitwidth in the $i$-th substructure on the task loss, it is crucial to recompute the corresponding step size for that bit setting.

In our approach, the step size at $i$-th substructure with $b$ bit is determined by minimizing the mean squared error (MSE) between the quantized and original values for both weights and activations. For activations, we further enhance stability by using a moving average of the step size computed at each batch of the calibration dataset. This reduces the dependence of the batch data and yields a more stable evaluation of $\Delta\mathcal{L}$.

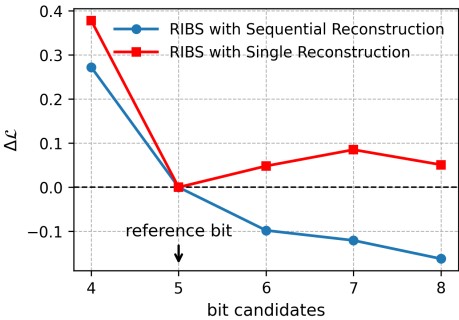

**Integration with reconstruction**    RIBS can integrate with the reconstruction methods, as outlined in Alg.1. In conventional reconstruction methods (Li et al., 2021; Wei et al., 2022), reconstruction proceeds sequentially from the first block to the last. As a result, later blocks are strongly influenced by earlier ones, leading to strong correlations between blocks, which is far from the independent assumption in Corollary 2. This implies that naively estimating the loss variation by changing the bit-width of a single block and then minimizing its reconstruction error introduces a significant mismatch between true and estimated variations. Indeed, as shown in Fig. 3, even when the

Figure 3: Loss variation across bit candidates with 5-bit reference point in the first block of MobileNetV2.

bit width of a block becomes increased and its representational capacity should improve, the task loss becomes increase, yielding $\Delta\mathcal{L}$. Consequently, minimizing the reconstruction error only for the modified block does not provide a reliable estimate of the true change in task loss.

To address this issue, we optimize the reconstruction error sequentially for all blocks following the corresponding block. This adjustment ensures that the estimate more faithfully reflects the true change in task loss. We can see that in Fig. 3, increasing the bit-width of a block results in $\Delta\mathcal{L} < 0$, consistent with the expected behavior.

---

**Algorithm 1** RIBS: Repeated ILP for Bit Selection

---

**Input**: $N$ substructures, candidate bits $\mathcal{B} = \{n^{(1)}, \ldots, n^{(K)}\}$, update size $M$, iterations $N_{iter}$
**Output**: Optimal bit configuration $\mathbf{b}^*$

1: Initialize reference configuration $\mathbf{b}^0$ with all 8-bit and step sizes
2: **for** iter = 1 to $N_{iter}$ **do**
3:     Randomly select $M$ substructures
4:     **for** each selected substructure $i$ **do**
5:       **for** each bit $n^{(k)} \in \mathcal{B}$ **do**
6:         Replace with $n^{(k)}$, and update update step size at $i$-th substructure
7:         **if** Reconstruction mode **then**
8:           Reconstruct with new configuration $\mathbf{b}_i^0 \equiv [b_1^0, \ldots, b_{i-1}^0, n^{(k)}, b_{i+1}^0, \ldots, b_N^0]$
9:         **end if**
10:         Compute loss $\mathcal{L}(\mathbf{b}_i^0)$ and $\Delta\mathcal{L}_{i,k}$ on calibration data
11:       **end for**
12:     **end for**
13:     Solve ILP in equation 3 to update $\mathbf{b}^0$
14: **end for**
15: **return** $\mathbf{b}^0$ as $\mathbf{b}^*$

---

## 4 EXPERIMENTS

We implement our method in PyTorch and conduct experiments on the ImageNet dataset (Deng et al., 2009) using ResNet-18 (He et al., 2016b), MobileNetV2 (Howard et al., 2017), and DeiT-T (Touvron et al., 2021). All pre-trained models are obtained from the timm library (tim). We evaluate the effect of mixed precision in both PTQ and QAT settings.

**Inplementation details**    We use 1024 calibration samples for bit selection and set $N_{iter} = 10$. The initial reference bit for all substructures is 8, and the bit candidate set is $\mathcal{B} \in \{2, \ldots, 8\}$. Task loss on unlabeled data is evaluated using distillation loss (temperature = 1) between the FP and quantized networks. To satisfy the resource constraint, we search all substructures, i.e. $M = N$ in the first iteration, followed by random $M$-substructure updates in later iterations. For BOP constraints, weights and activations are assigned the same bit-width. In RIBS with reconstruction under PTQ, we use 200 iterations for ResNet-18 and 1000 iterations for MobileNetV2 within each block reconstruction.

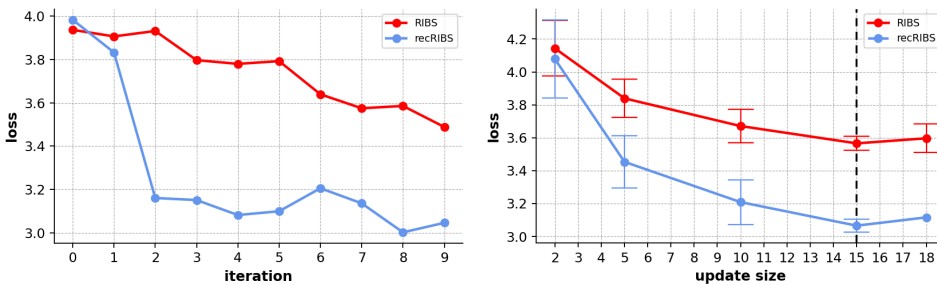

Figure 4: Task loss vs. iterations with update size 15 (left) and best task loss vs. number of update blocks (right) on MobileNetV2.

## 4.1 ABLATION STUDIES

**Iteration dependence** We investigate the behavior of the task loss for calibration data as a function of iteration. As Corollary 3 points out, the task loss is guaranteed to decrease when $M = 1$. However, with a large number of updates, due to the mismatch between true and ILP differences, the loss is not guaranteed to decrease. Nevertheless, as shown in Fig.4, we find that task loss generally decreases with increasing iterations in both RIBS and RIBS with reconstruction (recRIBS). **Number of update blocks at each step** We investigate how the best task loss in $N_{iter} = 10$ on calibration data varies with the update size $M$. As $M$ increases, the search space expands, which helps to avoid getting trapped in the local minima. However, this also amplifies the discrepancy between the true task loss difference and the ILP-estimated difference. Therefore, there exists a sweet spot for the update size, as illustrated in the right panel of Fig. 4. Based on this analysis for each model, we set $M = 5$ on Resnet-18, $M = 15$ for MobileNetV2, and $M = 28$ on DeiT-T.

## 4.2 EVALUATION IN PTQ

We conduct the comparison in single-precision and mixed-precision PTQ methods on ResNet-18, MobileNetV2, and DeiT-T. PTQ methods can be categorized into those with or without reconstruction. The (vanilla) RIBS searches the optimal bit configuration for the task loss without reconstruction, whereas RIBS with reconstruction searches the optimal bit configuration for the task loss with reconstruction. Table. 1 demonstrates that RIBS achieves a significantly better trade-off by a large margin between different model sizes and BOPs.

**CNN** Vanilla RIBS outperforms the baseline method that determines only the step size via MSE optimization (MSEinit), achieving up to $26.8\%$ improvement on 4/4-bit MobileNetV2, as shown in Table 1 (b). In recRIBS, we employ QDROP (Wei et al., 2022), a well-established reconstruction method for CNNs. As shown in Table 1(a) and (b), our approach surpasses the state-of-the-art (SOTA) single-precision methods. Specifically, recRIBS achieves $0.98\% - 1.6\%$ accuracy improvements over 4/4-bit QDrop on ResNet-18 and MobileNetV2, while requiring smaller model sizes and fewer BOPs. Furthermore, our method consistently outperforms the SOTA mixed-precision methods under the same or smaller model size and BOPs. In particular, recRIBS surpasses the recent method EMQ by up to $1.2\%$ improvement.

**ViT** In the DeiT experiments, we employ an improved version of RepQViT, an established PTQ method without reconstruction for ViTs. RepQViT adopts min–max quantization with two key techniques: (i) scale reparameterization, which converts channel-wise quantization of post-LayerNorm activations into layer-wise quantization with only a slight accuracy drop, and (ii) the use of $\log \sqrt{2}$ quantizers for post-softmax activations to better capture their non-uniform distribution.

We extend RepQViT by determining the step size via MSE optimization instead of min-max quantization, and by replacing $\log \sqrt{2}$ quantizer with the nuLSQ quantizer (Gongyo et al., 2024), which more effectively captures their non-uniform distribution. We refer to this enhanced variant as RepQViT+. In RIBS, we adopt RepQViT+ as the baseline: During bit selection, we employ channel-wise quantization for post-LayerNorm activations and the nuLSQ quantizer for post-softmax activations, after which RepQViT+ quantization is applied in the selected configuration.

Table 1: Comparison of single- and mixed-precision PTQ methods on ResNet-18, MobileNetV2, and DeiT-T. Methods: OMPQ (Ma et al., 2023a), EMQ (Dong et al., 2023), QDROP (Wei et al., 2022), HAWQ-V3 (Yao et al., 2021), BRECQ (Li et al., 2021), and MRECG (Ma et al., 2023b). MSEinit, RepQViT (Li et al., 2023), and RepQViT+ are directly compared with RIBS, while others are compared with recRIBS. † uses 8-bit for the first/last layers, ∗ denotes our implementation. "MP": mixed precision, "MS": model size (MiB), "BOPs": bit operations (G), "Top-1": accuracy (%).

(a) Results on ResNet-18

| Accuracy vs. model size | | | | Accuracy vs. BOPs | | | |
|---|---|---|---|---|---|---|---|
| Method | W/A | MS | Top-1 | Method | W/A | BOPs | Top-1 |
| FP32 | 32/32 | 44.6 | 72.54 | FP32 | 32/32 | 1858 | 72.54 |
| QDROP† | 4/4 | 5.81 | 69.62 | MSEinit∗ | 4/4 | 29.0 | 52.0 |
| OMPQ† | MP/4 | 5.5 | 69.38 | RIBS (ours)∗ | MP/MP | 28.5 | **53.36** |
| EMQ† | MP/4 | 5.5 | 70.12 | QDROP∗ | 4/4 | 29.0 | 69.13 |
| recRIBS (ours)∗† | MP/4 | 5.5 | **70.60** | MRECG | 4/4 | 29.0 | 69.46 |
| BRECQ† | MP/8 | 4.0 | 68.82 | recRIBS (ours)∗ | MP/MP | 28.5 | **70.20** |
| OMPQ† | MP/8 | 4.0 | 69.41 | MSEinit∗ | 6/6 | 65.3 | 71.61 |
| EMQ† | MP/8 | 4.0 | 69.92 | HAWQ-V3† | MP/MP | 72 | 70.22 |
| recRIBS (ours)∗† | MP/8 | 4.0 | **70.74** | RIBS (ours)∗ | MP/MP | 64.8 | **71.73** |

(b) Results on MobileNetV2

| Accuracy vs. model size | | | | Accuracy vs. BOPs | | | |
|---|---|---|---|---|---|---|---|
| Method | W/A | MS | Top-1 | Method | W/A | BOPs | Top-1 |
| FP32 | 32/32 | 13.4 | 72.89 | FP32 | 32/32 | 307 | 72.89 |
| BRECQ | MP/8 | 1.3 | 68.99 | MSEinit∗ | 4/4 | 4.8 | 13.81 |
| OMPQ | MP/8 | 1.3 | 69.62 | RIBS (ours)∗ | MP/MP | 4.8 | **40.59** |
| EMQ | MP/8 | 1.3 | 70.72 | QDROP∗ | 4/4 | 4.8 | 66.77 |
| recRIBS (ours) | MP/8 | 1.3 | **70.97** | recRIBS (ours)∗ | MP/MP | 4.8 | **68.36** |
| BRECQ | MP/8 | 1.5 | 70.28 | recRIBS (ours)∗ | MP/MP | 5.36 | **69.36** |
| EMQ | MP/8 | 1.5 | 70.75 | | | | |
| recRIBS (ours) | MP/8 | 1.5 | **71.99** | | | | |

(c) Results on DeiT-T

| Accuracy vs. BOPs | | | |
|---|---|---|---|
| Method | W/A | BOPs (G) | Top-1 |
| FP32 | 32/32 | 1284 | 72.13 |
| RepQViT+∗ | 6/6 | 45.13 | 71.12 |
| RIBS (ours)∗ | MP/MP | 45.13 | **71.21** |
| RepQViT+∗ | 4/4 | 20.06 | 54.32 |
| RIBS (ours)∗ | MP/MP | 20.01 | **59.94** |
| RepQViT† | 4/4 | 21.46 | 57.43 |
| RepQViT+∗† | 4/4 | 21.46 | 59.08 |
| RIBS (ours)∗ | MP/MP | 21.39 | **62.15** |
| RepQViT+∗ | 3/3 | 11.28 | 12.72 |
| RIBS (ours)∗ | MP/MP | 11.28 | **20.58** |
| RepQViT+∗† | 3/3 | 12.88 | 21.02 |
| RIBS (ours)∗ | MP/MP | 12.85 | **32.64** |

As shown in Table 1(c), we show the comparison under BOPs constraints corresponding to below 3-, 4-, and 6-bits. RIBS consistently outperforms RepQViT+ under these lower BOPs constraints with improvements of up to 11.6%.

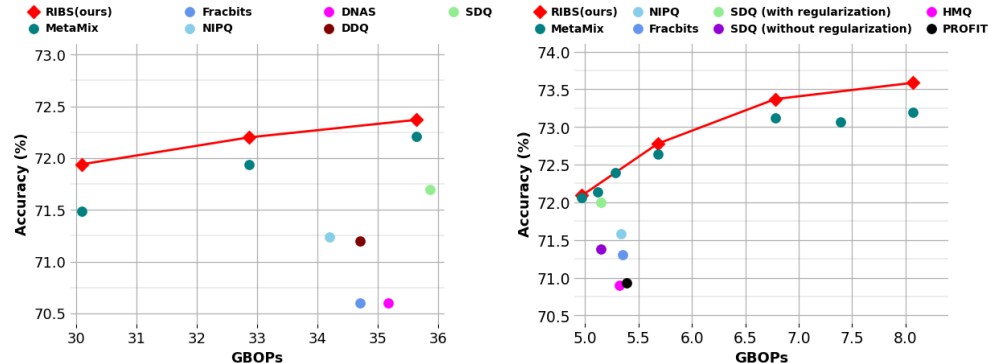

Figure 5: Top-1 accuracy vs. BOPs on ResNet-18 (left) and MobileNetV2 (right). Compared methods: FracBits-PACT (Yang & Jin, 2021), HAWQ-v3 (Yao et al., 2021), PROFIT (Park & Yoo, 2020), DNAS (Wu et al., 2018), DDQ (Zhang et al., 2021), NIPQ (Shin et al., 2023), SDQ (Huang et al., 2022), and MetaMix (Kim et al., 2024). † indicates 8-bit input activations in the first layer.

### 4.3 EVALUATION IN QAT

We evaluate mixed-precision QAT on ResNet-18 and MobileNetV2 under a wide range of BOP constraints. In these experiments, after the first iteration of layer-wise bit selection using the calibration data, we perform QAT with the full labeled dataset using LSQ (Esser et al., 2019), a well-established QAT method, on the selected bit configuration. Following the strategy of MetaMix (Kim et al., 2024), the current SOTA mixed-precision approach in QAT, we use ResNet-101 as the teacher network for ResNet-18, and MobilenetV2 with a $1.2\times$ width multiplier as the techer network for the original MobilenetV2. As shown in Fig.5, RIBS consistently outperforms existing mixed-precision methods.

## 5 CONCLUSION

In this paper, we propose RIBS, a novel bit-selection method. By repeatedly solving an ILP problem for bit selection with a limited number of update blocks, RIBS mitigates the gap between the true loss variation and its proxy, thereby yielding more optimal bit configuration for calibration data, and leading to improved performance.

To validate the effectiveness of RIBS, we conducted comprehensive experiments and comparative evaluations. We compared the performance of RIBS with SOTA single-precision and mixed-precision quantization approaches on both PTQ and QAT settings. Our evaluation covered three network architectures on the ImageNet dataset under a variety of model size and BOP constraints. The results demonstrate that RIBS consistently outperforms single-precision methods as well as existing SOTA mixed-precision approaches, highlighting its potential for advancing model compression.

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

## A  SENSITIVITY OF QUANTIZATION ON RESNET18

We show the quantization sensitivy in ResNet-18 in Fig.6. Similar to the case of MobileNetV2, the sensitivity significantly varies across layers.

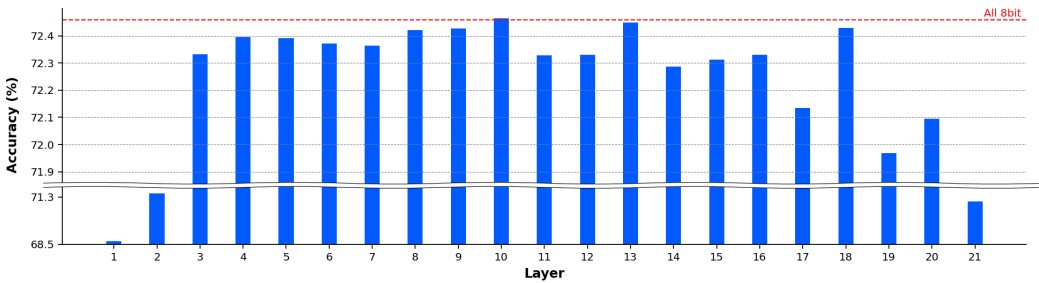

Figure 6: Sensitivity of quantization on each layer on RenNet-18. Each layer is quantized to 3 bits, while others are set to 8 bits.

## B  PROOF OF THEOREM AND COROLLARY

### B.1  PROOF OF THEOREM1

The condition of Theorem 1:

$$\left| \mathcal{L}(\mathbf{b}) - \sum_{i=1}^{N} \mathcal{L}(\mathbf{b}_{\hat{i}}^{0}) - \left( \mathcal{L}(\mathbf{b}^{*}) - \sum_{i=1}^{N} \mathcal{L}(\mathbf{b}_{\hat{i}}^{*0}) \right) \right| \leq \delta, \tag{8}$$

$$\sum_{i=1}^{N} \Delta\mathcal{L}(b_i) \geq \min_{\mathbf{b} \in \mathcal{B}^N} \sum_{i=1}^{N} \Delta\mathcal{L}(b_i) + 2\delta \quad \text{for} \quad \mathbf{b} \neq \mathbf{b}^{*}. \tag{9}$$

We define $f(\mathbf{b})$ as

$$f(\mathbf{b}) := \sum_{i=1}^{N} \Delta\mathcal{L}(b_i), \qquad \Delta\mathcal{L}(b_i) = \mathcal{L}(\mathbf{b}_{\hat{i}}^{0}) - \mathcal{L}(\mathbf{b}^{(0)}). \tag{10}$$

Let $\mathbf{b}^* \in \mathcal{B}$ be an optimal solution of the ILP that minimizes $f$ over $\mathcal{B}$. Define the interaction (non-additivity) error

$$e(\mathbf{b}) := \left(\mathcal{L}(\mathbf{b}) - \mathcal{L}(\mathbf{b}^{(0)})\right) - f(\mathbf{b}). \tag{11}$$

Using the notation, the first condition is expressed as

$$|e(\mathbf{b}) - e(\mathbf{b}^*)| = \left| \mathcal{L}(\mathbf{b}) - \sum_{i=1}^{N} \mathcal{L}(\mathbf{b}_{\hat{i}}^0) - \left( \mathcal{L}(\mathbf{b}^*) - \sum_{i=1}^{N} \mathcal{L}(\mathbf{b}_{\hat{i}}^{*0}) \right) \right| \leq \delta. \tag{12}$$

For $\delta > 0$ we have following conditions for all $\mathbf{b} \in \mathcal{B}$:

$$\left| e(\mathbf{b}) - e(\mathbf{b}^*) \right| \leq \delta, \tag{13}$$

$$f(\mathbf{b}) \geq f(\mathbf{b}^*) + 2\delta \quad \text{for all } \mathbf{b} \in \mathcal{B} \setminus \{\mathbf{b}^*\}. \tag{14}$$

By using

$$\mathcal{L}(\mathbf{b}) - \mathcal{L}(\mathbf{b}^*) = \left[ f(\mathbf{b}) - f(\mathbf{b}^*) \right] + \left[ e(\mathbf{b}) - e(\mathbf{b}^*) \right],$$

$f(\mathbf{b}) - f(\mathbf{b}^*) \geq 2\delta$, and equation 13, $e(\mathbf{b}) - e(\mathbf{b}^*) \geq -\delta$,

$$\mathcal{L}(\mathbf{b}) - \mathcal{L}(\mathbf{b}^*) \geq 2\delta + (-\delta) = \delta > 0.$$

Therefore $\mathcal{L}(\mathbf{b}) > \mathcal{L}(\mathbf{b}^*)$ for all $\mathbf{b} \neq \mathbf{b}^*$, proving that $\mathbf{b}^*$ is the minimizer of $\mathcal{L}$.

## B.2 PROOF OF COROLLARY 2

When all substructures are independent, $\mathcal{L}(\mathbf{b}) = \sum_i L_i(b_i)$ is satisfied. This leads to

$$
\begin{aligned}
\mathcal{L}(\mathbf{b}) - \sum_{i=1}^{N} \mathcal{L}(\mathbf{b}_{\hat{i}}^0) - \left( \mathcal{L}(\mathbf{b}^*) - \sum_{i=1}^{N} \mathcal{L}(\mathbf{b}_{\hat{i}}^{*0}) \right) &= \sum_i L_i(b_i) - \sum_i \left( L_i(b_i) + \sum_{j \neq i} L_j(b_j^0) \right) \\
&\quad - \left( \sum_i L_i(b_i^*) - \sum_i \left( L_i(b_i^*) + \sum_{j \neq i} L_j(b_j^0) \right) \right) \\
&= 0.
\end{aligned}
\tag{15}
$$

Therefore $\delta = 0$ and equation 9 is satisfied by definition.

## B.3 PROOF OF COROLLARY 3

We assume that the search space $\mathcal{B}^N$ is restricted to configurations satisfying $d_H(\mathbf{b}, \mathbf{b}^0) = 1$. Define the search space as $\mathcal{B}_0^N$. In this special case, all $b \in \mathcal{B}_0^N$ are expressed as $\mathbf{b}_{\hat{j}}^0$ for some $j$. By definition,

$$b^* = \arg\min_{\mathbf{b} \in \mathcal{B}_0^N} \Delta\mathcal{L}(b_i) = \arg\min_{\mathbf{b} \in \mathcal{B}_0^N} \mathcal{L}(\mathbf{b}) \tag{16}$$

is satisfied. This can alternatively be proved by following Theorem 1:

$$\mathcal{L}(\mathbf{b}) - \sum_{i=1}^{N} \mathcal{L}(\mathbf{b}_{\hat{i}}^0) = 0, \tag{17}$$

and

$$\left| \mathcal{L}(\mathbf{b}) - \sum_{i=1}^{N} \mathcal{L}(\mathbf{b}_{\hat{i}}^0) - \left( \mathcal{L}(\mathbf{b}^*) - \sum_{i=1}^{N} \mathcal{L}(\mathbf{b}_{\hat{i}}^{*0}) \right) \right| = 0. \tag{18}$$

Thus, $\delta = 0$ and equation 9 is satisfied by definition.

