# OpenReview forum: "Repeated Integer Linear Programming for Bit Selection in neural network quantization"
_ICLR.cc/2026/Conference — ICLR 2026 Conference Withdrawn Submission_

### Official Review · Reviewer_2uoc · 2025-10-26

**Soundness:** 2
**Presentation:** 3
**Contribution:** 3
**Rating:** 4
**Confidence:** 5

**Summary:**

This paper proposes RIBS (Repeated Integer Linear Programming for Bit Selection), a novel framework for determining optimal bit-width configurations in mixed-precision quantization of neural networks. The key insight is to cast the bit selection process as a series of integer linear programming (ILP) subproblems, which are iteratively solved to efficiently navigate the combinatorial configuration space. The approach is flexible and can be seamlessly integrated with both post-training quantization (PTQ) and quantization-aware training (QAT) pipelines, requiring only a small amount of unlabeled calibration data. Extensive experiments on standard benchmarks—including ResNet-18, MobileNetV2, and DeiT-T models evaluated on ImageNet—demonstrate that RIBS consistently achieves superior accuracy–efficiency trade-offs compared to existing uniform-precision and mixed-precision baselines.

**Strengths:**

- The introduction of a repeated ILP formulation for bit selection represents a novel and insightful perspective on the combinatorial optimization challenge inherent in mixed-precision quantization. The overall workflow of RIBS is clearly articulated and conceptually straightforward, while maintaining a relatively low computational overhead. These characteristics make the approach easy to reproduce and appealing for both research exploration and practical deployment.
- The RIBS framework exhibits strong flexibility and broad applicability. It is largely quantization-method agnostic, allowing seamless integration with a variety of both post-training quantization (PTQ) and quantization-aware training (QAT) pipelines. This generality enhances its potential impact, as the method can serve as a modular optimization component across diverse quantization strategies and architectures.

**Weaknesses:**

- The paper validates RIBS only on the ImageNet classification benchmark using three moderate-sized models. Its scalability to larger architectures (e.g., vision or language foundation models) and generalizability to other tasks (such as object detection, segmentation, or NLP) remain unexplored. This omission limits the scope and external validity of the claims.
- In the PTQ evaluation, the full-precision baselines vary across compared methods, which may significantly influence the post-quantization accuracy and confound comparisons. Similarly, in the QAT setting, training configurations and costs do not appear to be consistently controlled. For instance, the authors employ a specialized distillation strategy not adopted by previous baselines, and fail to report key hyperparameters (e.g., number of epochs, learning rate schedules), which are crucial for assessing fairness and training efficiency.
- Although the overall framework is conceptually simple, its practical performance depends strongly on several hyperparameters—such as the update size (M), number of iterations (N₍iter₎), and the definition of "substructures". The paper would benefit from a more systematic discussion or analysis on how these hyperparameters should be tuned for different architectures and tasks, and how sensitive the final quantization accuracy is to these design choices. Moreover, because substructures are randomly sampled in each iteration, the final accuracy is expected to exhibit variance across runs. Yet, only a single result is reported. It remains unclear whether this value corresponds to the best run, the average, or a typical outcome across multiple trials.
- Since the central objective of RIBS is to achieve an optimal balance between computational efficiency and model accuracy, the paper should include a more thorough analysis of time and resource overhead. In particular, comparisons against both metric-based and optimization-based mixed-precision methods in terms of computational cost, convergence time, and accuracy would help clarify RIBS’s true efficiency. Given that the method requires tuning several model-specific hyperparameters, a transparent discussion of its end-to-end computational footprint is essential for assessing practical usability.

**Questions:**

- Could the authors provide additional evidence or analysis to substantiate that the proposed method indeed yields a better—or provably optimal—solution under the conditions stated in Theorem 1? A more rigorous theoretical or empirical justification would strengthen the claimed optimality of the approach.
- In the quantization-aware training (QAT) experiments, was the proposed method used solely to determine the optimal bit-width configuration, or were the quantized weights obtained from RIBS also employed as initialization for subsequent QAT fine-tuning?
- It would be valuable to visualize or analyze how the bit-width assignments differ across layers when using the proposed method versus a baseline approach. Such comparisons could offer deeper insight into what structural patterns RIBS captures and how they contribute to performance gains.
- As noted in the weaknesses, the proposed algorithm appears largely independent of the specific loss function adopted in Algorithm 1. However, the paper lacks experiments demonstrating robustness with respect to different loss formulations. Including such results would reinforce the generality and stability of the method.

Overall, my primary concern lies in the experimental section, which appears too limited to substantiate the central claim that the proposed method effectively bridges the gap between the ILP-derived solution and the true minimizer. The current empirical evidence does not convincingly demonstrate that the approach consistently yields near-optimal or superior results in practice. Additional controlled experiments or analyses—such as comparisons with ground-truth optimal solutions on smaller test cases or convergence studies—would be necessary to validate this theoretical claim.

---

### Official Review · Reviewer_zKij · 2025-10-26

**Soundness:** 3
**Presentation:** 2
**Contribution:** 3
**Rating:** 4
**Confidence:** 5

**Summary:**

The paper proposes RIBS which approximates mixed-precision quantization by iteratively exploring the bit-configuration space through solving an integer linear programs. It requires a small set of calibration data and can be applicable to both PTQ and QAT.

**Strengths:**

1. Practical problem to solve: Sensitivity difference to quantization bit-width leads to per-layer different bit-width quantization aiming to extreme optimization. However, decision of proper per-layer bit-width in quantization is an important and complex problem with balancing tradeoff between efficiency and accuracy. While the model can have n^l complexity (n for the number of bit-width candidate and l for the number of layers), for example, 3^53 =1.938×10^25 with [2,4,8]-bits in MobileNet-v2, finding the best case in vast combination of bit-widths efficiently is in high demand but difficult to solve.
2. Strong results across all budget (BOPs and model size) on both PTQ and QAT. Considering this paper is proposing only bit-width selection method and the saturated accuracy improvement of recent quantization works, improvements shown in the paper is valid.

**Weaknesses:**

I integrated the weaknesses points into Section:Qeustions.

**Questions:**

1. In Corollary 2 and 3, is assumption of independent loss that divides and adds the contributions of each layer separately (= the bit-width of each substructure affects only its own loss contribution) with the hamming distance of one a valid method to calculate correct L(b)? While the ILP can be constructed with this assumption, in the realistic case, application of quantization on earlier layers always affects to later layers due to numerical difference of output activations from unquantized earlier layers. For example, as stated in the paper about strong correlations, when we quantize layer j to extremely lower bit, it will also affect the loss contribution of later layers (j+1, j+2, …). This would be also same on block-wise update strategy.
2. In Figure 5, seems the number of computation (BOPs) of RIBS exactly matches to the BOPs of MetaMix. What is the selected bit-width in each cases? Doesn’t it match to the bit-width of MetaMix? If it matches, how RIBS can give better result than MetaMix?
3. While the sensitivity of quantization on each layer is displayed, the bit-width selection result cannot be shown on the paper. What is the result of the bit-width selection of RIBS on both MobileNet-v2 and ResNet-18? Does it match to the per-layer sensitivity the authors proposed?
4. Thorough explanation of the notations are required. What does ‘K’, ‘Cmax’, and superscript of ‘b’ stand for in Equation 2, 3, and 4?
5. When we lower the target bit-width below 4-bit, accuracy degradation seems large compared to the unquantized network which can lead to the problem of severe damage in 2-bit and 3-bit. How’s the appearance of 2, 3-bit on each models? In what layers, 2-bit and 3-bit appears?
6. Is there any reason you didn’t include 1-bit in the bit candidate set?
7. Is there any reason you didn’t compare MobileNet-v3 rather than MobileNet-v2?
8. How much is the searching cost (time) of RIBS in both PTQ and QAT? In QAT, do we run RIBS bit-width searching every iteration or just only once?

Minor:
1. Dagger sign in Figure 5 doesn’t appear on the figure.
2. The quality of some figures (low resolution and incomplete appearance of some characters) is not good.

---

### Official Review · Reviewer_sMKD · 2025-10-29

**Soundness:** 2
**Presentation:** 3
**Contribution:** 3
**Rating:** 4
**Confidence:** 5

**Summary:**

This paper explores bit allocation across model layers for mixed-precision quantization. Moving from existing ILP-based bit allocation approach, the paper proposes a repeated ILP scheme, where the model is first quantized to a higher precision, then redo ILP with the quantized model to a lower precision. Theoreitical justifications are provided for the proposed method. Additional techniques like random block update are utilized to further enhance the performance.

**Strengths:**

1. Novelty-wise, this paper identifies the previously overlooked issue of inaccurate proxy of single-round ILP method for bit allocation. Theoreitical justifications are provided to show the condition for ILP to work.
2. The presentation of the paper is clear and easy to follow
3. Experiments across different models and quantization methods show improved performance with the bit allocation achieved by the proposed method.

**Weaknesses:**

1. The cost of the proposed method is not discussed. As shown in the algorithm, the proposed method requires multiple rounds of bit allocation and weight reconstruction before achieving the target precision. In the case where the target precision is low or the model is deep requiring more times of sampling, the cost may quickly scale up and even go beyond the QAT cost. Given the larger cost, the method may need to be compared with training-based bit allocation methods like [1,2,3]
2. There are multiple things in the proposed method that may contribute to the improved performance. It is unsure if the improved performance is contributed by better quantization bit allocation, better initialization point for PTQ/QAT from the iterative search process, or both. Ablation study on training a model with the final bit allocation from scratch or perform PTQ to the bit allocation directly on pretrained FP model is needed to better understand the process.
3. Experiments are limited to small-scale models, making it unsure if the method can scale up well. Since the method depends on sampling a small portion of the model for each step of update, having a deeper model may leads to exponential growth of sampling needed to cover the full model.

[1] BSQ, ICLR 2021, https://arxiv.org/abs/2102.10462

[2] CSQ, DAC 2023, https://arxiv.org/abs/2212.02770

[3] MSQ, ICCV 2025, https://arxiv.org/abs/2507.22349v1

**Questions:**

Please provide more discussion regarding the weakness 1 and 2 listed above.

---

### Official Review · Reviewer_nhL4 · 2025-10-30

**Soundness:** 2
**Presentation:** 3
**Contribution:** 2
**Rating:** 4
**Confidence:** 4

**Summary:**

The paper introduces Repeated Integer Linear Programming for Bit Selection (RIBS), an efficient, iterative method to solve the mixed-precision quantization problem. The method approximates the complex combinatorial optimization problem of mixed-precision quantization as a sequence of solvable ILP problems. RIBS is computationally light, uses a small calibration dataset , and is compatible with both PTQ and QAT methods. It achieves SOTA accuracy in both PTQ (as recRIBS) and QAT settings across various CNNs and ViTs.

**Strengths:**

The method leverages a small set of unlabeled calibration samples, making it efficient to find the solutions.

The method is for both QAT and PTQ which is method-unaware. And The experimental results are highly significant. It achieves SOTA accuracy in both QAT and PTQ settings.

**Weaknesses:**

The mathematical analysis: The analysis is driven by the observation that layer sensitivity to bit-width is non-uniform. Achieving the best bit-width combination hinges on accurately modeling the loss increase, $\Delta\mathcal{L}(b_i)$, for each layer $i$ when its precision is changed5. Yet, in the resulting ILP formulation, this essential per-layer loss term is used as a fixed, constant coefficient in the objective function, which potentially limits its ability to capture true loss dynamics across configurations.

Unquantified Computational Overhead in recRIBS: The paper claims low computational overhead. However, the core technical innovation in recRIBS—sequentially optimizing the reconstruction error for all blocks following the modified block when calculating $\Delta \mathcal{L}$ (Algorithm 1, line 8) —is a potentially time-consuming operation.

Motivation：The motivation for mixed-precision quantization is the observed variation in accuracy degradation across layers. If this degradation isn't solely based on a layer's position (layer number), why isn't it correlated with more objective metrics like the number of parameters or computational size (MACs) of that specific layer?

**Questions:**

Please look at the weaknesses above.

---

### Note · Authors · 2025-12-02

**Comment:**

Thank you very much for all the reviewers’ comments. We need additional time to revise our paper, so we have decided to withdraw it from this conference and plan to resubmit it to another conference.

**Withdrawal Confirmation:**

I have read and agree with the venue's withdrawal policy on behalf of myself and my co-authors.